# Risk factors for hospitalisation in Welsh infants with a congenital anomaly

Peter S Y Ho  ,[1] Maria A Quigley,[1,2] David F Tucker,[3] Jennifer J Kurinczuk[1,2]

[1]National Perinatal Epidemiology Unit, Nuffield Department of Population Health, University of Oxford, Oxford, UK
[2]National Institute for Health Research (NIHR) Policy Research Unit- Maternal Health and Care, National Perinatal Epidemiology Unit, Nuffield Department of Population Health, University of Oxford, Oxford, UK
[3]Public Health Wales, Public Health Knowledge & Research, Congenital Anomaly Register & Information Service for Wales, Public Health Wales, Swansea, UK

**Correspondence to**
Dr Peter S Y Ho; Peterhosy@hotmail.com

## ABSTRACT

**Objectives** To investigate risk factor associated with hospitalisation of infants with a congenital anomaly in Wales, UK.

**Design** A population-based cohort study.

**Setting** Data from the Welsh Congenital Anomaly Register and Information Service linked to the Patient Episode Database for Wales and livebirths and deaths from the Office for National Statistics.

**Patients** All livebirths between 1999 and 2015 with a diagnosis of a congenital anomaly, which was defined as a structural, metabolic, endocrine or genetic defect, as well as rare diseases of hereditary origin.

**Main outcome measures** Adjusted OR (aOR) associated with 1 or 2+ hospital admissions in infancy versus no admissions were estimated for sociodemographic, maternal and infant factors using multinomial logistic regression for the subgroups of all, isolated, multiple and cardiovascular anomalies.

**Results** 25 523 infants affected by congenital anomalies experienced a total of 50 705 admissions in infancy. Risk factors for ≥2 admissions were younger maternal age ≤24 years (aOR: 1.17; 95% CI 1.06 to 1.30), maternal smoking (aOR: 1.20; 1.10 to 1.31), preterm birth (aOR: 2.52; 2.25 to 2.83) and moderately severe congenital heart defects (aOR: 6.25; 4.47 to 8.74). Girls had an overall decreased risk of 2+ admissions (aOR: 0.84; 0.78 to 0.91). Preterm birth was a significant risk factor for admissions in all anomaly subgroups but the effect of the other characteristics varied according to anomaly subgroup.

**Conclusions** Over two-thirds of infants with an anomaly are admitted to hospital during infancy. Our findings identified sociodemographic and clinical characteristics contributing to an increased risk of hospitalisation of infants with congenital anomalies.

## INTRODUCTION

Congenital anomalies are a leading cause of infant morbidity and hospitalisation.[1] Infants who are born with a major congenital anomaly such as a congenital heart defect (CHD), Down syndrome and congenital diaphragmatic hernia often experience anomaly-related sequelae and complications, and require more frequent hospitalisation compared with children without an anomaly.

There is some evidence that infant characteristics can influence hospitalisation of children affected by specific anomalies. For example, previous studies have shown that

### What is known about the subject?

► Congenital anomalies are a leading cause of infant morbidity and hospitalisation.
► Evidence about the risk factors contributing to the hospitalisation of infants with congenital anomalies is limited.

### What this study adds?

► Infants who had ≥2 admissions were more likely to be boys, born preterm, born with a moderately severe congenital heart defect, or have mothers who were younger and smokers, compared to those with ≤1 admission.
► Preterm birth is the strongest risk factor for hospitalisation in infants with congenital anomalies or rare diseases of congenital origin.
► The mechanisms by which maternal age, smoking and infant sex are risk factors for hospitalisation are similar for anomaly-specific admissions and admissions due to other causes.

preterm birth, low birth weight, disease severity and comorbidity are significantly associated with an increased risk of excess hospitalisation of infants with spina bifida, choanal atresia, congenital heart defects, orofacial clefts, congenital diaphragmatic hernia, biliary atresia, gastroschisis and Down syndrome[2–8]; while the effects of sociodemographic and maternal are less well established. Further research on risk factors for hospitalisation, particularly those associated with a wider range of congenital anomalies are needed to inform planning of healthcare and social interventions aimed at reducing infant morbidity and hospitalisation.

We aimed to investigate risk factors for hospitalisation in the first year after birth among infants with congenital anomalies.

## METHODS
### Study design and record linkage

A national population-based cohort study was established using registry data from the

Congenital Anomaly Register and Information Service (CARIS) for Wales, linked to Patient Episode Database for Wales (PEDW), and births and deaths registration data from the Office for National Statistics (ONS),[9–11] and deidentified for analysis.

Deterministic and probabilistic linkage between datasets were performed using an unique Anonymised Linkage Field (ALF) encrypted code provided by NHS Wales Informatics Service, based on infant's NHS number, name, date of birth and address. All infants in the CARIS and PEDW datasets have a validated ALF and hence each infant was matched with hospital admissions using one to multiple merge; unmatched infants were treated as zero admissions. The outputs of linkage were validated by the Secure Anonymised Information Linkage (SAIL) team.

## Study population

The inclusion criteria for this study were:

1. All livebirths between 1999 and 2015 with birth weight ≥500 g, gestational age ≥$22^{+0}$ weeks, with a confirmed or probable diagnosis of a congenital anomaly reported to CARIS. CARIS defines congenital anomalies as structural, metabolic, endocrine or genetic defects, as well as rare diseases of hereditary origin present in the child or fetus at the end of pregnancy, even if not detected until after birth.[9] Therefore, the 10th revision of International Classification of Diseases and Related Health Problems (ICD-10) 'Q', 'P35' and 'P37' codes, as well as other non-'Q' ICD-10 codes of congenital anomalies and rare diseases were also included in this study.

2. All hospital admissions (ie, emergency, elective and transfer admissions) of infants with a congenital anomaly. An admission is defined as a hospital stay of a patient using a bed on premises controlled by the one healthcare provider.[12] An admission is ended by a discharge, transfer to another NHS provider or death. All infants who were admitted from birth due to medical reasons were included in the analysis.

Exclusion criteria:

1. Admissions solely for the birth of babies (ie, birth hospitalisation) were excluded.

## Categorisation of anomaly

Congenital anomalies were categorised according to the European Surveillance of Congenital Anomalies (EUROCAT) subgroup classification[13]; other rare diseases which are not included in the EUROCAT subgroup classification were categorised according to the ICD chapter headings[14] (online supplemental table 1).

Infants were considered as having an isolated congenital anomaly (or disease) if a single anomaly (or disease) was diagnosed and reported to CARIS. Infants were considered as having multiple anomalies (or diseases) if more than one anomaly (or diseases) was diagnosed, either within the same body system or involving different body systems. Infants who were diagnosed with a syndrome involving more than one anomaly (or disease) were considered as having multiple anomalies (or diseases).

## Outcome

The outcome was total hospital admissions for any cause in infancy (period from birth up to the first birthday) and infants were categorised as having 0, 1 or 2+ admissions.

## Statistical analysis

Multinomial logistic regression was conducted to estimate the adjusted OR (aOR) of hospital admission and the corresponding 95% CI according to the categories of sociodemographic, maternal and infant factors, comparing infants who had 1 or 2+ admissions versus those who had no admission (the comparison group). Any hospital transfers were treated as separate admissions in order to assess the full burden and utilisation of hospitalisation.

The modelling strategy aimed to identify the strongest risk factors for hospital admission. Variables included a priori in the multivariable models were infant sex, gestational age at birth and disease severity for cardiovascular anomalies as these are known determinants of infant morbidity and hospitalisation,[5 15–18] and these variables were included in the final model regardless of their statistical significance in the univariable analysis. The remaining variables that were associated with hospitalisation in univariable analysis (p<0.1) were also initially included in multivariable models to observe their effect size and statistical significance and they were removed from the multivariable model if no category of the variable improved the fit of the data (ie, p<0.05). As birth weight and gestational age at birth were highly collinear (correlation coefficient r=0.75), only gestational age at birth was included in the final model due to its greater clinical utility. All multivariable analysis was carried out separately for the following subgroups: infants with any anomaly; infants with isolated anomalies; infants with multiple anomalies and infants with cardiovascular anomalies.

A further subgroup analysis was conducted for those babies who had at least one admission to investigate whether risk factors were different between infants with anomaly-specific admissions and those whose admissions were for 'other-causes'. Here, anomaly-specific admissions were defined as hospitalisation of infants due to 'congenital anomalies, deformations and chromosomal abnormalities' (ICD-10 codes: Q00-Q99/chapter XVII) as the primary reason for admission. Crude ORs with corresponding 95% CIs were estimated for each explanatory variable comparing these two groups of infants. Significant variables from the univariable analysis (p<0.1) were included in the multivariable analysis. P<0.05 was considered as statistically significant in all final models.

A sensitivity analysis was conducted by adding year of birth to the final model for each subgroup to investigate if the effect of the key risk factors changed.

Missing data for the majority of variables were few and they are not shown due to the risk of statistical disclosure. All analyses were performed using complete case analysis. All statistical analyses were performed using Stata V.13.[19]

## Patient and public involvement

All proposals to use data within the SAIL Databank are subject to review by an independent Information Governance Review Panel (IGRP) for privacy risk, data governance and public benefit assessment. The IGRP is made up of a range of independent experts as well as members of the public.

## RESULTS

Between 1999 and 2015, 25 523 babies were born alive and affected by congenital anomalies or rare disease of congenital origin, and required a total of 50 705 admissions in infancy. Overall, approximately a quarter of infants had only one admission, over 40% had two or more admissions and more than one-third had no admission in infancy. Among infants with isolated anomalies, the largest number of admissions occurred in infants with a cardiovascular anomaly (8% of all admissions of babies with an anomaly).

Table 1 shows that infants who had two or more admissions compared with those who had one or no admission were more likely to live in a deprived area (30.6% vs 28.6% or 26.7%), to have mothers who were younger (33.3% vs 29.3% or 28.2%), nulliparous (45.2% vs 43.6% or 43.3%), and smokers (30.7% vs 27.1% or 25.5%), to be boys (61.2% vs 59.6% or 59.9%), born preterm (21.3% vs 14.0% or 9.1%), to have a low birth weight (21.5% vs 13.5% or 10.1%), or to be one of a multiple pregnancy (3.9% vs 2.9% or 2.4%).

In the univariable analysis, social deprivation, maternal age, parity, maternal smoking and preterm birth were significantly associated with an increased or decreased risk for repeated admissions for 'all', 'all isolated' and 'all multiple' anomaly subgroups (table 2). Except for social deprivation and maternal smoking, the same variables (ie, maternal age and preterm births) which were associated with an increased risk for repeated admissions were also consistently associated with an increased or decreased risk of single admissions of infants for these anomaly subgroups.

Tables 3 and 4 show the adjusted OR for variables significantly associated with (p<0.05) the number of admissions in infancy by different anomaly or rare disease of congenital origin subgroups. Preterm birth was consistently the strongest risk factor associated with increased 1 and 2+ admissions of infants across all anomalies subgroups with an overall 2½-fold increased risk of 2+ admissions (aOR: 2.52; 95% CI 2.25 to 2.83), and a 68% increased risk of a single admission (aOR: 1.68; 1.47 to 1.92) compared with term infants. Preterm infants with a cardiovascular anomaly had the highest risk of repeated hospital admissions (aOR: 3.76; 2.86 to 4.93) compared

with term infants with a cardiovascular anomaly. Infant sex was a significant factor for 2+ admissions of infants across all anomaly subgroups. For example, girls with an anomaly had an overall 16% decreased risk of 2+ admissions, compared with boys (aOR: 0.84; 0.78 to 0.91); and girls with a cardiovascular anomaly had a 25% decreased risk of 2+ admissions, compared with boys (aOR: 0.75; 0.62 to 0.91). In addition, infants with the most severe and moderately severe CHD had a 2.7 (aOR: 2.72; 1.58 to 4.71) and 6.3 (aOR: 6.25; 4.47 to 8.74) folds increased risk of 2+ admissions, respectively compared with infants with a less severe CHD. Infants with the most severe CHD had 61% reduced risk of being admitted at all (aOR: 0.39; 0.16 to 0.96). For infants with multiple anomalies, those whose mothers were aged ≤24 years had a 22% increased risk of 2+ admissions (aOR: 1.22; 1.06 to 1.41), while those whose mothers were aged ≥35 had a 46% decreased risk of 2+ admissions (aOR: 0.54; 0.47 to 0.63) and a 42% decreased risk of single admissions (aOR: 0.58; 0.48 to 0.69) compared with those whose mothers were aged 25–29 years. Maternal smoking was associated with an increased risk of 2+ admissions for all anomalies (aOR: 1.20; 1.10 to 1.31) and all isolated anomalies (aOR: 1.27; 1.13 to 1.41). The effects of each variable did not vary substantially when adjusted for year of birth (online supplemental table 2).

Among infants who had at least one admission, maternal age, parity, maternal smoking and infant sex were not associated with having admissions for causes related to the anomaly as opposed to admissions for other causes (online supplemental tables 3,4). However, preterm infants were less like to have anomaly-specific admissions (aOR: 0.47; 0.42 to 0.52), whereas infants with mothers having a history of anomalies in previous pregnancies were more likely to have anomaly-specific admissions (aOR: 1.26; 1.11 to 1.43).

## DISCUSSION

Using linked deidentified data from CARIS, PEDW and ONS, we investigated a comprehensive range of risk factors associated with hospital admissions during infancy in babies born with congenital anomalies or rare diseases of congenital origin. Preterm birth was the strongest risk factor associated with an increased risk of one and two or more infant hospitalisation across all anomaly or rare disease of congenital origin subgroups. In addition, infants with two or more admissions had a higher risk profile than those with only one or no admission. Infants who had two or more admissions were more likely to be boys, born preterm, or to have mothers who were younger and smokers or for those who were born with a moderately severe CHD anomaly, compared with those with one or no admission.

Previous studies have shown that infants with a congenital anomaly were associated with a significantly high risk of hospitalisation compared with infants without an anomaly in the general population due to anomaly-related

**Table 1** Factors associated with hospitalisation of infants with any anomaly or rare disease of congenital origin

| | Number of admissions | | |
| | 2+ | 1 | 0 |
| | No. of infants (%) | No. of infants (%) | No. of infants (%) |
| Characteristics | n=10785 | n=6043 | n=8695 |
|---|---|---|---|
| Sociodemographic factors | | | |
| Townsend quintile | | | |
| 1 (least deprived) | 1542 (14.3) | 991 (16.4) | 1272 (16.3) |
| 2 | 1645 (15.3) | 956 (15.8) | 1316 (16.9) |
| 3 | 1928 (17.9) | 1102 (18.3) | 1571 (20.1) |
| 4 | 2351 (21.8) | 1257 (20.8) | 1558 (20.0) |
| 5 (most deprived) | 3296 (30.6) | 1727 (28.6) | 2080 (26.7) |
| Maternal ethnicity | | | |
| White | 6729 (94.5) | 3547 (93.8) | 4252 (95.0) |
| Other | 394 (5.5) | 236 (6.2) | 224 (5.0) |
| Maternal age at birth, years | | | |
| Mean (SD) (years) | 27.8 (6.3) | 28.3 (6.2) | 28.6 (6.2) |
| Median (IQR) | 28 (23–32) | 28 (24–33) | 29 (24–33) |
| ≤24 | 3596 (33.3) | 1768 (29.3) | 2204 (28.2) |
| 25–29 | 2880 (26.7) | 1697 (28.1) | 2096 (26.8) |
| 30–34 | 2585 (24.0) | 1526 (25.3) | 2046 (26.2) |
| ≥35 | 1720 (16.0) | 1050 (17.4) | 1468 (18.9) |
| Maternal factors | | | |
| Parity | | | |
| Nulliparous | 4572 (45.2) | 2454 (43.6) | 3029 (43.3) |
| ≥1 | 5535 (54.8) | 3173 (56.4) | 3973 (56.7) |
| Multiple fetus | | | |
| Yes | 417 (3.9) | 175 (2.9) | 185 (2.4) |
| No | 10365 (96.1) | 5866 (97.1) | 7640 (97.6) |
| Maternal smoking | | | |
| Smoker | 2200 (30.7) | 1036 (27.1) | 1136 (25.5) |
| Non/ex-smoker | 4958 (69.3) | 2783 (72.9) | 3315 (74.5) |
| Anomalies in previous pregnancies | | | |
| Yes | 928 (12.0) | 506 (12.2) | 582 (12.0) |
| No | 6811 (88.0) | 3637 (87.8) | 4280 (88.0) |
| Infant factors | | | |
| Infant sex | | | |
| Male | 6595 (61.2) | 3599 (59.6) | 5205 (59.9) |
| Female | 4187 (38.8) | 2441 (40.4) | 3486 (40.1) |
| Birth weight, g | | | |
| Mean (SD) | 3024 (810) | 3207 (697) | 3286 (660) |
| Median (IQR) | 3140 (2610–3570) | 3280 (2860–3660) | 3350 (2950–3710) |
| <2500 (low birth weight) | 2311 (21.5) | 815 (13.5) | 806 (10.1) |
| ≥2500 | 8441 (78.5) | 5209 (86.4) | 7166 (89.9) |
| Gestational age, weeks | | | |
| Mean (SD) | 37.9 (3.3) | 38.6 (2.6) | 39.0 (2.3) |
| Median (IQR) | 39 (37–40) | 39 (38–40) | 39 (38–40) |
| <37 (preterm) | 2287 (21.3) | 844 (14.0) | 727 (9.1) |
| ≥37 (term) | 8465 (78.7) | 5178 (86.0) | 7249 (90.9) |

Missing data are not shown due to risk of statistical disclosure.

**Table 2** Univariable analyses: factors associated with repeated (2+) and single (1) admissions by anomaly/rare disease subgroup

| Factors | 2+ admissions | | | 1 admission | | |
|---|---|---|---|---|---|---|
| | Unadjusted OR† | | | Unadjusted OR† | | |
| | All | Isolated | Multiple | All | Isolated | Multiple |
| **Townsend quintile** | | | | | | |
| 1 | 1 (reference) | 1 (reference) | 1 (reference) | 1 (reference) | 1 (reference) | 1 (reference) |
| 2 | 1.03 (0.93 to 1.14) | 1.04 (0.92 to 1.18) | 1.06 (0.88 to 1.28) | 0.93 (0.83 to 1.05) | 0.88 (0.77 to 1.01) | 1.09 (0.87 to 1.37) |
| 3 | 1.01 (0.92 to 1.12) | 1.06 (0.93 to 1.20) | 0.93 (0.78 to 1.10) | 0.90 (0.81 to 1.01) | 0.88 (0.77 to 1.01) | 0.92 (0.74 to 1.14) |
| 4 | 1.24 (1.13 to 1.37) | 1.23 (1.08 to 1.39) | 1.24 (1.04 to 1.47) | 1.04 (0.93 to 1.16) | 1.02 (0.89 to 1.16) | 1.08 (0.87 to 1.34) |
| 5 | 1.31 (1.19 to 1.43) | 1.32 (1.18 to 1.48) | 1.28 (1.08 to 1.51) | 1.07 (0.96 to 1.18) | 1.02 (0.90 to 1.15) | 1.17 (0.96 to 1.44) |
| **Maternal ethnicity** | | | | | | |
| White | 1 (reference) | 1 (reference) | 1 (reference) | 1 (reference) | 1 (reference) | 1 (reference) |
| Other | 1.11 (0.94 to 1.32) | 1.10 (0.88 to 1.38) | 1.06 (0.82 to 1.39) | 1.26 (1.05 to 1.52) | 1.34 (1.06 to 1.69) | 1.13 (0.82 to 1.56) |
| **Maternal age at delivery, years** | | | | | | |
| ≤24 | 1.19 (1.10 to 1.28) | 1.16 (1.05 to 1.27) | 1.23 (1.07 to 1.41) | 0.99 (0.91 to 1.09) | 0.97 (0.87 to 1.07) | 1.05 (0.88 to 1.25) |
| 25–29 | 1 (reference) | 1 (reference) | 1 (reference) | 1 (reference) | 1 (reference) | 1 (reference) |
| 30–34 | 0.92 (0.85 to 1.00) | 0.89 (0.81 to 0.98) | 0.94 (0.81 to 1.09) | 0.92 (0.84 to 1.01) | 0.93 (0.83 to 1.04) | 0.90 (0.75 to 1.07) |
| ≥35 | 0.53 (0.49 to 0.58) | 0.50 (0.45 to 0.56) | 0.55 (0.48 to 0.64) | 0.55 (0.50 to 0.61) | 0.54 (0.48 to 0.61) | 0.58 (0.48 to 0.69) |
| **Parity** | | | | | | |
| Nulliparous | 1 (reference) | 1 (reference) | 1 (reference) | 1 (reference) | 1 (reference) | 1 (reference) |
| ≥1 | 0.73 (0.69 to 0.77) | 0.73 (0.68 to 0.79) | 0.76 (0.68 to 0.84) | 0.78 (0.73 to 0.84) | 0.78 (0.72 to 0.85) | 0.80 (0.71 to 0.91) |
| **Multiple fetus** | | | | | | |
| Yes | 1.66 (1.39 to 1.98) | 1.92 (1.53 to 2.39) | 1.20 (0.89 to 1.60) | 1.23 (1.00 to 1.52) | 1.43 (1.11 to 1.84) | 0.87 (0.60 to 1.27) |
| No | 1 (reference) | 1 (reference) | 1 (reference) | 1 (reference) | 1 (reference) | 1 (reference) |
| **Maternal smoking** | | | | | | |
| Smoker | 1.30 (1.19 to 1.41) | 1.32 (1.19 to 1.48) | 1.24 (1.09 to 1.43) | 1.09 (0.99 to 1.20) | 1.06 (0.94 to 1.20) | 1.14 (0.96 to 1.34) |
| Non/ex-smoker | 1 (reference) | 1 (reference) | 1 (reference) | 1 (reference) | 1 (reference) | 1 (reference) |
| **Anomalies in previous pregnancies** | | | | | | |
| Yes | 1.00 (0.89 to 1.11) | 1.04 (0.91 to 1.20) | 0.81 (0.67 to 0.99) | 0.98 (0.86 to 1.11) | 1.00 (0.86 to 1.16) | 0.90 (0.70 to 1.14) |
| No | 1 (reference) | 1 (reference) | 1 (reference) | 1 (reference) | 1 (reference) | 1 (reference) |
| **Infant sex*** | | | | | | |
| Male | 1 (reference) | 1 (reference) | 1 (reference) | 1 (reference) | 1 (reference) | 1 (reference) |
| Female | 0.95 (0.90 to 1.00) | 0.94 (0.87 to 1.01) | 0.88 (0.79 to 0.97) | 1.01 (0.95 to 1.08) | 1.07 (0.99 to 1.16) | 0.86 (0.76 to 0.97) |
| **Gestational age,* weeks** | | | | | | |
| <37 | 2.95 (2.70 to 3.23) | 3.25 (2.89 to 3.64) | 2.15 (1.87 to 2.49) | 1.79 (1.61 to 1.98) | 1.91 (1.67 to 2.18) | 1.48 (1.24 to 1.77) |
| ≥37 | 1 (reference) | 1 (reference) | 1 (reference) | 1 (reference) | 1 (reference) | 1 (reference) |

*Variable identified for inclusion a priori.
†For 'All' anomaly subgroup, adjusted variables include maternal age at delivery, maternal smoking, infant sex and gestational age; for 'isolate' anomaly subgroup, adjusted variables include maternal smoking, infant sex and gestation age; for 'multiple' anomaly subgroup, adjusted variables include maternal age at delivery, infant sex and gestational age. Statistical significance at p<0.05.

**Table 3**  Multivariable analyses: factors associated with repeated (2+) and single (1) admissions by anomaly or rare disease of congenital origin subgroup

| | 2+ admissions | | | 1 admission | | |
| | Adjusted OR† | | | Adjusted OR† | | |
| Factors | All | Isolated | Multiple | All | Isolated | Multiple |
| --- | --- | --- | --- | --- | --- | --- |
| **Maternal age at delivery, years** | | | | | | |
| ≤24 | 1.17 (1.06 to 1.30) | | 1.22 (1.06 to 1.41) | 1.00 (0.89 to 1.12) | | 1.04 (0.88 to 1.24) |
| 25–29 | 1 (reference) | | 1 (reference) | 1 (reference) | | 1 (reference) |
| 30–34 | 0.99 (0.89 to 1.10) | | 0.94 (0.81 to 1.09) | 0.94 (0.83 to 1.06) | | 0.90 (0.75 to 1.07) |
| ≥35 | 0.93 (0.82 to 1.04) | | 0.54 (0.47 to 0.63) | 0.95 (0.83 to 1.09) | | 0.58 (0.48 to 0.69) |
| **Maternal smoking** | | | | | | |
| Smoker | 1.20 (1.10 to 1.31) | 1.27 (1.13 to 1.41) | | 1.06 (0.96 to 1.17) | 1.04 (0.92 to 1.18) | |
| Non/ex-smoker | 1 (reference) | 1 (reference) | | 1 (reference) | 1 (reference) | |
| **Infant sex*** | | | | | | |
| Male | 1 (reference) | 1 (reference) | 1 (reference) | 1 (reference) | 1 (reference) | 1 (reference) |
| Female | 0.84 (0.78 to 0.91) | 0.82 (0.74 to 0.90) | 0.90 (0.81 to 1.00) | 0.95 (0.87 to 1.04) | 1.01 (0.91 to 1.12) | 0.87 (0.77 to 0.99) |
| **Gestational age,* weeks** | | | | | | |
| <37 | 2.52 (2.25 to 2.83) | 2.83 (2.41 to 3.31) | 2.19 (1.90 to 2.53) | 1.68 (1.47 to 1.92) | 1.85 (1.55 to 2.21) | 1.50 (1.26 to 1.79) |
| ≥37 | 1 (reference) | 1 (reference) | 1 (reference) | 1 (reference) | 1 (reference) | 1 (reference) |

*Variable identified for inclusion a priori.

†For 'All' anomaly subgroup, adjusted variables include maternal age at delivery, maternal smoking, infant sex and gestational age; for 'isolate' anomaly subgroup, adjusted variables include maternal smoking, infant sex and gestation age; for 'multiple' anomaly subgroup, adjusted variables include maternal age at delivery, infant sex and gestational age. Statistical significance at p<0.05.

comorbidities and complications, which require ongoing inpatient treatment and investigations.[6 20–22]

Maternal and infant factors have an impact on hospitalisation in infant with congenital anomalies through different potential mechanisms. For example, previous studies have shown that teenage mothers may have an increased risk of adverse neonatal outcomes due to inadequate antenatal care and social support, and other behavioural determinants such as maternal smoking, alcohol and recreational drug misuse compared with older mothers.[23] Infants born to older mothers tend to have a reduced risk of unintentional injuries, an increased uptake of routine immunisation in childhood and may receive better parenting.[24] It has been shown that maternal smoking during pregnancy increases the risk of respiratory and infectious diseases in infants[25–27]; possible mechanisms include intrauterine stress and impaired immune system due to nicotine substances and tobacco smoking, although the exact biological mechanism is not clearly defined.[28] In our subgroup analysis, we found no significant difference in the relationship between maternal age and anomaly-specific admissions or other-cause admissions, or those between maternal smoking and anomaly-specific admissions or other-cause admissions, which suggests that the mechanisms by which maternal age and smoking are risk factors for hospitalisation of infants with an anomaly are similar for anomaly-specific admissions and admissions due to other causes.

Preterm birth has consistently been shown to be associated with an increased risk of infant morbidity and hospitalisation; preterm infants generally have a lower physiological reserve and a higher susceptibility to comorbidities and infections compared with term infants.[29] In addition, clinicians may be more cautious and adopt a lower threshold for admitting preterm infants with a congenital anomaly when complications arise.[30] Previous twin studies have shown that boys have an increased risk of respiratory distress syndrome, intrauterine growth restriction and overall morbidities compared with girls and therefore are more likely to require hospitalisation.[16 31–33] Our subgroup analysis suggests that many infants with congenital anomalies who were born preterm required hospitalisation for conditions related to prematurity rather than the congenital anomaly, although it is possible that clinicians tend to ascribe 'prematurity' as the primary cause of admission for preterm babies regardless of their anomaly status.

Infants born with the most severe CHD (eg, single ventricle and hypoplastic heart syndromes) often experience life-threatening events at birth and hence the mortality risk among these infants is considerably higher during their first admission compared with infants with less severe CHD.[34 35] On the other hand, infants born with a moderately severe CHD are now more likely to survive due to advances in diagnostics, and medical and surgical interventions[36]; however, they are also likely to experience long-term morbidities and complications,[5 37 38] which may explain the highest risk of repeated hospitalisation in this group of infants.

The risk factors for increased hospitalisation in infants with congenital anomalies found in this study are similar to known risk factors for hospitalisation in the general

**Table 4**  Unadjusted and adjusted factors associated with repeated (2+) and single (1) admissions of infants with cardiovascular anomalies

| Factors | 2+ admissions | | 1 admission | |
|---|---|---|---|---|
| | Cardiovascular, n=997 | | | |
| | Unadjusted OR† | Adjusted OR‡ | Unadjusted OR† | Adjusted OR‡ |
| Townsend quintile | | | | |
| 1 | 1 (reference) | | 1 (reference) | |
| 2 | 1.11 (0.80 to 1.53) | | 0.80 (0.56 to 1.12) | |
| 3 | 1.35 (0.99 to 1.85) | | 0.94 (0.68 to 1.31) | |
| 4 | 1.10 (0.81 to 1.50) | | 1.05 (0.77 to 1.43) | |
| 5 | 1.40 (1.05 to 1.85) | | 0.97 (0.72 to 1.30) | |
| Maternal ethnicity | | | | |
| White | 1 (reference) | | 1 (reference) | |
| Other | 0.82 (0.49 to 1.38) | | 0.91 (0.53 to 1.57) | |
| Maternal age at delivery, years | | | | |
| ≤24 | 1.20 (0.95 to 1.53) | | 0.94 (0.72 to 1.23) | |
| 25–29 | 1 (reference) | | 1 (reference) | |
| 30–34 | 1.02 (0.80 to 1.31) | | 1.08 (0.83 to 1.41) | |
| ≥35 | 0.57 (0.43 to 0.75) | | 0.67 (0.51 to 0.90) | |
| Parity | | | | |
| Nulliparous | 1 (reference) | | 1 (reference) | |
| ≥1 | 0.92 (0.76 to 1.11) | | 0.86 (0.70 to 1.05) | |
| Multiple fetus | | | | |
| Yes | 1.86 (1.15 to 3.00) | | 1.55 (0.91 to 2.62) | |
| No | 1 (reference) | | 1 (reference) | |
| Maternal smoking | | | | |
| Smoker | 1.39 (1.07 to 1.79) | | 1.07 (0.81 to 1.42) | |
| Non/ex-smoker | 1 (reference) | | 1 (reference) | |
| Anomalies in previous pregnancies | | | | |
| Yes | 1.32 (0.94 to 1.86) | | 1.20 (0.84 to 1.71) | |
| No | 1 (reference) | | 1 (reference) | |
| Infant sex* | | | | |
| Male | 1 (reference) | 1 (reference) | 1 (reference) | 1 (reference) |
| Female | 0.66 (0.55 to 0.79) | 0.75 (0.62 to 0.91) | 0.95 (0.78 to 1.15) | 0.95 (0.78 to 1.15) |
| Gestational age,* week | | | | |
| <37 | 3.33 (2.55 to 4.35) | 3.76 (2.86 to 4.93) | 1.97 (1.46 to 2.65) | 1.97 (1.46 to 2.66) |
| ≥37 | 1 (reference) | 1 (reference) | 1 (reference) | 1 (reference) |
| CHD disease severity* | | | | |
| Less severity | 1 (reference) | 1 (reference) | 1 (reference) | 1 (reference) |
| Moderate severity | 5.83 (4.19 to 8.11) | 6.25 (4.47 to 8.74) | 1.14 (0.75 to 1.75) | 1.18 (0.77 to 1.81) |
| Most severe | 2.67 (1.57 to 4.57) | 2.72 (1.58 to 4.71) | 0.38 (0.15 to 0.94) | 0.39 (0.16 to 0.96) |

*Variable identified for inclusion a priori.
†Statistically significant at p<0.1 for unadjusted OR.
‡Adjusted variables include infant sex, gestational age and disease severity. Statistically significant at p<0.05.
CHD, congenital heart defect.

paediatric population, in which previous studies have found that being a boy and maternal smoking are associated with preterm birth leading to increased risk of hospitalisation.[39 40]

Having a major anomaly increases the chance of an infant being born preterm,[15] and they may contribute to the same chain of event leading to hospitalisation.

The main strengths of this study include the national study population, robust study design and data collection. Infants with congenital anomalies or rare diseases of congenital origin were identified from CARIS, an active surveillance and multiple source reporting system covering all livebirths in Wales which maximises case finding and internal validity. The use of routinely collected data from PEDW is particularly advantageous for this hospitalisation study due to the large size of the dataset and long period of data collection. A further strength relates to the broad definition of congenital anomalies, which includes structural and chromosomal defects, as well as rare diseases of congenital origin such as metabolic, endocrine and congenital blood disorders. Consequently, we were able to investigate risk factors associated with hospitalisation of infants with a wide range of congenital anomalies and rare diseases of congenital origin.

There are several limitations. As PEDW is a hospital administrative dataset, data on admissions was not always complete which can be due to infants being treated in a specialist centre outside the catchment area of NHS Wales. In addition, the number of variables explored in this study was restricted by what was available in the dataset. It is possible that institutional factors such as outpatient follow-ups which were not available in this dataset are also important determinants of hospitalisation of infants with congenital anomalies.

## CONCLUSIONS

Infants with congenital anomalies and rare diseases of congenital origin have a high risk of hospitalisation in infancy. Our findings can help clinicians to identify infants who are likely to require hospitalisation according to their sociodemographic and clinical characteristics. Public health interventions such as promoting good prenatal care and smoking cessation may have an important role in reducing the need for hospitalisation in infants with a congenital anomaly.

**Acknowledgements** We thank the SAIL team for providing the datasets and their assistance with data linkage for this study, and the CARIS team for their support on the dataset.

**Contributors** PH contributed to the design of the study; acquisition, analysis, interpretation of data; drafting and revising the manuscript critically; approved the final version to be published; and agrees to be accountable for all aspects of the work in ensuring that questions related to the accuracy or integrity of any part of the work are appropriately investigated and resolved; and accepts full responsibility for the work, overall content and conduct of the study as the guarantor. MQ and JK contributed to the design of the study; acquisition, analysis and interpretation of data for the work; supervised the project and revised the manuscript critically; approved the final version to be published; and agree to be accountable for all aspects of the work in ensuring that questions related to the accuracy or integrity of any part of the work are appropriately investigated and resolved. DT contributed to the acquisition, interpretation of data for the study; revised the manuscript critically; approved the final version to be published; and agrees to be accountable for all aspects of the work in ensuring that questions related to the accuracy or integrity of any part of the work are appropriately investigated and resolved.

**Funding** This study is part-funded by the National Institute for Health Research (NIHR) Policy Research Programme, conducted through the NIHR Policy Research Unit in Maternal Health and Care, 108/0001. The views expressed are those of the author(s) and not necessarily those of the NIHR or the Department of Health and Social Care. PH is funded by a Clarendon—Green Templeton College—Nuffield Department of Population Health Scholarship, University of Oxford.

**Competing interests** None declared.

**Patient consent for publication** Not applicable.

**Ethics approval** This study does not involve human participants. The secondary anonymised health data for this study was provided by the SAIL Databank in Wales. This study was approved by the SAIL Information Governance Review Panel (IGRP) and did not require separate research ethics committee review. All statistical disclosure control and checks were strictly followed according to the SAIL Databank and ONS guidelines, to ensure that individual infants were not deductively identifiable.

**Provenance and peer review** Not commissioned; externally peer reviewed.

**Data availability statement** All data relevant to the study are included in the article or uploaded as supplementary information. All data relevant to the study are included in the article or uploaded as supplementary information. The secondary anonymised health data for this study was provided by the Secure Anonymised Information Linkage (SAIL) Databank in Wales.

**ORCID iD**
Peter S Y Ho http://orcid.org/0000-0003-4962-6808

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
