## [Reviewer comments · BMJ Paediatrics Open]

ARTICLE DETAILS

TITLE (PROVISIONAL)	Risk factors for hospitalisation in Welsh infants with a congenital anomaly
AUTHORS	Ho, Peter Quigley, Maria Tucker, David Kurinczuk, Jenny

VERSION 1 – REVIEW

REVIEWER	Reviewer name: Dr. Jeremy Miles Institution and Country: Google Inc, 340 Main St, Venice, California, 90291, United States Competing interests: None
REVIEW RETURNED	29-Aug-2021

GENERAL COMMENTS	(Statistical Review) Removing predictors (even those which are not statistically significant) from the model can bias other estimates; this is potentially justified when the sample size is small, as additional predictors can cause a loss of power, but with this large sample size, it seems unnecessary. The authors are essentially doing stepwise regression, with its well known problems. The results section talks about RRRs (and adjusted RRRs) but the statistical analysis section describes logistic regression - logistic regression gives odds ratios, not relative risk ratios. How were relative risk ratios (and their CIs) calculated? The data sharing statement tells us how the authors obtained the data, but it is not clear how someone wishing to replicate the results would obtain the same data. I believe that I found the dataset at: https://data.ukserp.ac.uk/Asset/View/7 - but the author analysis uses data only up to 2015, when 2019 data are available. (If I had access to the data I would have calculated RRR and CIs.)
---

REVIEWER	Reviewer name: Dr. Deborah A. Ridout Institution and Country: University College London Institute of Child Health, United Kingdom of Great Britain and Northern Ireland Competing interests: None
REVIEW RETURNED	27-Aug-2021

GENERAL COMMENTS	This is an interesting study which links retrospective data from the Welsh Congenital Anomaly Register and Information Service (CARIS) to hospital administrative data and investigates risk factors associated with hospitalisations. Further details and justifications for some of the methodological approaches would improve the manuscript. • The study uses data from Wales and this should be included in the
--

	title.  • No detail is provided about the linkage process, which factors were used to match records? What was the success of the matching? Were there any unmatched patients in CARIS and / or hospitalisation data? How did the authors distinguish between unmatched CARIS patients and zero admissions for example? • The rationale for the development of the 'risk' models has not been explained fully and it seems a hybrid stepwise, apriori clinical importance approach has been taken. What is the objective of fitting the models – is it to develop a prognostic model or to understand the importance of risk factors, this will influence the motivation for producing a parsimonious model. The hybrid approach is confusing and not justified or explained clearly. I would caution against including only factors in a multivariable model that reach a certain level of significance. I recommend including all variables based on clinical importance and then examine the size of effects, as well exploring the degree of attenuation between univariable and multivariable models. • Did the authors investigate any temporal trends in the data? • In general, the presentation of the tables is clear. Table 2 should indicate somewhere the RRR is relative to the risk in the zero admission patients. Also, I do not understand the footnote $p < 0.1$, what is this referring to? • Table 4: Is the unadjusted column based on univariable analysis?
--	---

REVIEWER	Reviewer name: Dr. Krishna Kishore Umapathi Institution and Country: Rush University Medical Center, United States Competing interests: none
REVIEW RETURNED	15-Sep-2021

GENERAL COMMENTS	Thank you for allowing me to review the article titled "RISK FACTORS FOR HOSPITALISATION IN INFANTS WITH A CONGENITAL ANOMALY" by Ho et al. Although the authors have performed and documented a study which is good methodologically, I don't think there is added advantage or knowledge by adding this study to existent literature. The comorbidities that the authors mention lead to admissions are well known risk factors with or without the presence forr a congenital anomaly
---

VERSION 1 – AUTHOR RESPONSE

25.10.2021

Dear Editor,

Thanks for your reviews and comments. Please find the responses as follows. Please find the relevant revisions in the manuscript and supp. information in red.

Peter

Editor in Chief Comments to Author

Title add "Wales"

Discussion delete the 1st sentence. Journal policy is to AVOID describing a study as the first. This is for others to decide following publication.

Respond in full to the comments of the reviewers

Response: revised as suggested.

Associate Editor

Comments to the Author:

Two reviewers have noted similar concerns with the variable selection approach in the multivariable analyses; these concerns appear straightforward to address.

One reviewer notes that the risk factors for increased hospitalisation in children with congenital anomalies were similar to known risk factors for hospitalisation in the general population; the discussion could address this observation.

Response: Please find the discussions added to the Discussion section.

Reviewer: 1

Dr. Deborah A. Ridout, University College London

Comments to the Author

This is an interesting study which links retrospective data from the Welsh Congenital Anomaly Register and Information Service (CARIS) to hospital administrative data and investigates risk factors associated with hospitalisations. Further details and justifications for some of the methodological approaches would improve the manuscript.

- The study uses data from Wales and this should be included in the title.

Response: revised in title.

- No detail is provided about the linkage process, which factors were used to match records? What was the success of the matching? Were there any unmatched patients in CARIS and / or hospitalisation data? How did the authors distinguish between unmatched CARIS patients and zero admissions for example?

Response: We have added more detail on this in the Methods section: Deterministic and probabilistic linkage between datasets were performed using a unique Anonymised Linkage Field (ALF) encrypted code provided by NHS Wales Informatics Service, based on infant's NHS number, name, date of birth and address. All infants in the CARIS and PEDW datasets have a validated ALF and hence each infant was matched with hospital admissions using one to multiple merge; unmatched infants were treated as zero admissions. The outputs of linkage were validated by the Secure Anonymised Information Linkage (SAIL) team.

- The rationale for the development of the 'risk' models has not been explained fully and it seems a hybrid stepwise, apriori clinical importance approach has been taken. What is the objective of fitting the models – is it to develop a prognostic model or to understand the importance of risk factors, this will influence the motivation for producing a parsimonious model. The hybrid approach is confusing and not justified or explained clearly. I would caution against including only factors in a multivariable model that reach a certain level of significance. I recommend including all variables based on clinical importance and then examine the size of effects, as well exploring the degree of attenuation between univariable and multivariable models.

Response: The results presented in this paper are part of a wider study which also looked at risk factors for infant mortality in the same population. A modelling strategy was developed that was consistent across all outcomes (mortality, hospitalisation) and subgroups (from all infants with congenital anomalies, to those in particular anomaly groups such as cardiovascular anomalies). The sample size was relatively small for some of these outcomes/subgroups and therefore we decided not to include all potential risk factors in the final multivariable models. We included a conservative cut-off for dropping variables from the multivariable models (based on a significant p-value for one category of a risk factor

in one of the two sets of RRR). We have now added more details in the revised version.

- Did the authors investigate any temporal trends in the data?

Response: We have now included year of birth in the final model as a sensitivity analysis (see supplementary Table 2). The effects of each variable did not change substantially when they were adjusted for year of birth.

- In general, the presentation of the tables is clear. Table 2 should indicate somewhere the RRR is relative to the risk in the zero admission patients. Also, I do not understand the footnote $p < 0.1$, what is this referring to?

Response: Please see revision in table 4.

- Table 4: Is the unadjusted column based on univariable analysis?

Response: yes, it is.

Reviewer: 2

Dr. Jeremy Miles, Google Inc

Comments to the Author

(Statistical Review)

Removing predictors (even those which are not statistically significant) from the model can bias other estimates; this is potentially justified when the sample size is small, as additional predictors can cause a loss of power, but with this large sample size, it seems unnecessary. The authors are essentially doing stepwise regression, with its well known problems.

Response: please see response to reviewer 1 and the revised Methods statistical analysis.

The results section talks about RRRs (and adjusted RRRs) but the statistical analysis section describes logistic regression - logistic regression gives odds ratios, not relative risk ratios. How were relative risk ratios (and their CIs) calculated?

Response: As the outcome had three categories (0, 1, 2+ admissions), we used multinomial logistic regression rather than binary logistic regression. The resulting coefficients are relative risk ratios (RRR) rather than odds ratios.

The data sharing statement tells us how the authors obtained the data, but it is not clear how someone wishing to replicate the results would obtain the same data. I believe that I found the dataset at: <https://data.ukserp.ac.uk/Asset/View/7> - but the author analysis uses data only up to 2015, when 2019 data are available. (If I had access to the data I would have calculated RRR and CIs.)

Response: When the study was conducted in 2018, the request of the dataset from SAIL was up to 2017. However, it was noted that the available infant hospitalisation data beyond 2016 (birth cohort up to 2015) was not completed at the time of study and hence we decided not to include those incomplete data.

Reviewer: 3

Dr. Krishna Kishore Umapathi, Rush University Medical Center

Comments to the Author

Thank you for allowing me to review the article titled "RISK FACTORS FOR HOSPITALISATION IN INFANTS WITH A CONGENITAL ANOMALY" by Ho et al. Although the authors have performed and documented a study which is good methodologically, I don't think there is added advantage or knowledge by adding this study to existent literature. The comorbidities that the authors mention lead to admissions are well known risk factors with or without the presence for a congenital anomaly

VERSION 2 – REVIEW

REVIEWER	Reviewer name: Dr. Jeremy Miles
-----------------	---------------------------------

	Institution and Country: Google Inc, 340 Main St, Venice, California, 90291, United States Competing interests: None
REVIEW RETURNED	24-Nov-2021

GENERAL COMMENTS	I would like to thank the authors for addressing my comments. I am still unclear about one issue. Quoting from the authors' response to reviewers: Original review: The results section talks about RRRs (and adjusted RRRs) but the statistical analysis section describes logistic regression - logistic regression gives odds ratios, not relative risk ratios. The Response: As the outcome had three categories (0, 1, 2+ admissions), we used multinomial logistic regression rather than binary logistic regression. The resulting coefficients are relative risk ratios (RRR) rather than odds ratios. However, multinomial logistic regression is still logistic regression. It gives odds ratios, not risk ratios. Given an event, with probability p, the odd are given by: $p / (1 - p)$ The risk is simply p. So the odds ratio and the risk ratio are not interchangeable. (The estimates from multinomial logistic regression can be obtained by running multiple binary logistic regression analyses, with the reference and outcome - indeed this is what was recommended in the days before multinomial logistic regression software was available. Risk ratios can be obtained using Poisson regression, there is discussion in this CrossValidated answer: https://stats.stackexchange.com/questions/18595/poisson-regression-to-estimate-relative-risk-for-binary-outcomes
--

REVIEWER	Reviewer name: Dr. Deborah A. Ridout Institution and Country: University College London Institute of Child Health, United Kingdom of Great Britain and Northern Ireland Competing interests: None
REVIEW RETURNED	19-Nov-2021

GENERAL COMMENTS	The authors have addressed all my earlier comments and this has improved the manuscript which I would recommend for publication.
--

VERSION 2 – AUTHOR RESPONSE

30.11.2021

Response to Dr. Miles (reviewer 2): Thanks for your comments and the useful reference which we have reviewed. We adopted the terminology relative risk ratios (RRR) as this is what is used in the statistical output in Stata but agree that odd ratios (OR) is the correct term to interpret our findings. I have now changed the occurrences of RRR to OR in the manuscript and relevant supplementary tables.